# Tetrodotoxins in French Bivalve Mollusks—Analytical Methodology, Environmental Dynamics and Screening of Bacterial Strain Collections

**DOI:** 10.3390/toxins13110740

**Published:** 2021-10-20

**Authors:** Damien Réveillon, Véronique Savar, Estelle Schaefer, Julien Chevé, Marie-Pierre Halm-Lemeille, Dominique Hervio-Heath, Marie-Agnès Travers, Eric Abadie, Jean-Luc Rolland, Philipp Hess

**Affiliations:** 1Ifremer, DYNECO, Laboratoire Phycotoxines, F-44000 Nantes, France; Veronique.Savar@ifremer.fr (V.S.); Estelle.Schaefer@ifremer.fr (E.S.); 2Ifremer, LITTORAL, F-35800 Dinard, France; Julien.Cheve@ifremer.fr; 3Ifremer, LITTORAL, F-14520 Port en Bessin, France; Marie.Pierre.Halm.Lemeille@ifremer.fr; 4LEMAR, Université de Brest, Ifremer, CNRS, IRD, F-29280 Plouzané, France; Dominique.Hervio.Heath@ifremer.fr; 5Ifremer, SG2M, Laboratoire LSEM, F-29280 Plouzané, France; 6Ifremer, SG2M, Laboratoire LGPMM, F-17390 La Tremblade, France; Marie.Agnes.Travers@ifremer.fr; 7IHPE, Université de Montpellier, CNRS, Ifremer, Université de Perpignan Via Domitia, F-34000 Montpellier, France; Jean.Luc.Rolland@ifremer.fr; 8Ifremer, Biodivenv, F-97231 Le Robert, France; Eric.Abadie@ifremer.fr; 9MARBEC, Université de Montpellier, IRD, Ifremer, CNRS, F-34000 Montpellier, France

**Keywords:** emerging toxins, TTXs, REPHY, REMI, coastal and seafood contamination

## Abstract

Tetrodotoxins (TTXs) are potentially lethal paralytic toxins that have been identified in European shellfish over recent years. Risk assessment has suggested comparatively low levels (44 µg TTX-equivalent/kg) but stresses the lack of data on occurrence. Both bacteria and dinoflagellates were suggested as possible biogenic sources, either from an endogenous or exogenous origin. We thus investigated TTXs in (i) 98 shellfish samples and (ii) 122 bacterial strains, isolated from French environments. We optimized a method based on mass spectrometry, using a single extraction step followed by ultrafiltration without Solid Phase Extraction and matrix-matched calibration for both shellfish and bacterial matrix. Limits of detection and quantification were 6.3 and 12.5 µg/kg for shellfish and 5.0 and 10 µg/kg for bacterial matrix, respectively. Even though bacterial matrix resulted in signal enhancement, no TTX analog was detected in any strain. Bivalves (either *Crassostrea gigas* or *Ruditapes philippinarum*) were surveyed in six French production areas over 2.5–3 month periods (2018–2019). Concentrations of TTX ranged from ‘not detected’ to a maximum of 32 µg/kg (Bay of Brest, 17 June 2019), with events lasting 2 weeks at maximum. While these results are in line with previous studies, they provide new data of TTX occurrence and confirm that the link between bacteria, bivalves and TTX is complex.

## 1. Introduction

Tetrodotoxin (TTX) is a low molecular weight and polar neurotoxin that blocks the receptor site 1 of voltage-gated sodium channels [1]. For many years, it has mainly been associated with puffer fish-derived food poisoning in Asia, as reviewed by Guardone et al. [2]. Hitherto, TTX and its more than 30 known analogs (TTXs) have been found in an impressive diversity of phylogenetically unrelated organisms (both terrestrial, e.g., amphibians, and aquatic, e.g., fish from the Tetraodontidae family, worms, cephalopods, gastropods, bivalves, see for example the review of Noguchi and Arakawa [3]). A new TTX-like analogue (*m/z* 318.1) was even recently reported in the gastropod *Charonia lampas* from the Portuguese coast [4]. However, these toxins are considered an emerging threat in Europe and in bivalves (see reviews [5,6,7]).

The first TTX-related shellfish poisoning in Europe was in 2007, caused by a specimen of the gastropod *Charonia lampas* containing >300 mg TTX/kg in the digestive gland, and about three times more 5,6,11-trideoxyTTX [8]. Since then, several research groups have identified low levels of TTXs in bivalve mollusks, notably two simultaneously published studies in Greece and the UK [9,10]. These two studies showed the occurrence of TTXs in several species of bivalves in very diverse environments, i.e., the Mediterranean Sea and coastal areas of the English Channel (Southern UK), and triggered a temporary import ban in the Netherlands, a pivotal platform for shellfish trade in Europe, for shellfish with TTX concentrations exceeding the Dutch detection limit at the time (20 µg TTX/kg). Even though this trade ban was rapidly lifted as not compliant with EU internal market regulations, the need for new data and for risk assessment arose rapidly. In 2017, the European Food Safety Authority published its formal risk assessment [11], recommending a provisional limit in bivalves of 44 µg TTX-eq./kg fresh shellfish meat. This recommendation by EFSA clearly reflects a public health concern for shellfish consumers.

Official controls for paralytic shellfish toxins had been based on animal testing for decades worldwide [12,13], however, the initial validation of that methodology had focused on saxitoxins, a major group of paralyzing toxins, known to contaminate shellfish since the 1930s [14,15,16]. In addition, the traditional mouse bioassay is not an appropriate test for TTXs since it has a detection limit significantly above the level recommended by EFSA [17]. Since 1986, legislation is in place in Europe to abandon animal tests whenever non-animal alternatives are available [18]. As suggested by an earlier working group of the *European Centre for the Validation of Alternative Methods*, ECVAM [19], recent advances in the detection of saxitoxins by chemical analytical methods, and the formal interlaboratory validation of that methodology, have made it possible to replace the live animal test (mouse bioassay) by liquid chromatography coupled to fluorescence detection. The validated method selected for official control of saxitoxins (STXs), i.e., the Lawrence method [20], does unfortunately not detect TTXs. Therefore, a clear need exists for method validation for surveys and eventually for the official control of TTX(s) in shellfish. Many different methods were used to detect and quantify TTXs, e.g., reviewed by Bane et al. [7] but LC-MS/MS is recommended [11], and HILIC-MS/MS methods in particular are now considered a reference since they are both specific and comparatively easy to implement [21]. Still, this methodology is relatively recent and not yet interlaboratory validated for TTXs. Thus, method implementation and optimization still depend on available instruments and in-house protocols. Hence, the present study dedicated some efforts to method optimization for the quantitative detection of TTXs.

Depending on the organism considered, it is assumed that TTX can have either or both an endogenous (i.e., symbiotic bacteria) and an exogenous origin (i.e., from the diet) [7,22]. As for the producer of TTX(s), many reports point toward a bacterial primary source (reviewed by [3,23,24]), however, the biosynthetic pathway is still unknown despite valuable efforts [25]. According to Magarlamov and colleagues [24], more than 150 TTX-producing bacterial strains have been reported, dominated by the genera *Vibrio* and *Bacillus* that account for more than 30% and 15% of all strains, respectively. When focusing on bivalves, only two bacteria (*Vibrio cholerae* and *V. parahaemolyticus*) have been isolated from *M. edulis* and *C. gigas* and proven to contain TTX when cultivated for 10 out of the 11 strains [9]. The hypothesis of a link between the presence of *Vibrio* spp. and TTX in bivalves was tested in different studies [9,26,27,28,29,30] but does not yet appear to be confirmed. The hypothesis of bacterial production of TTXs accumulating in shellfish is also coherent with the risk profile determined by Turner’s group [30] for shellfish growing in shallow, estuarine areas of temperatures exceeding 15 °C and significant freshwater input.

Hence, this study was aimed at the detection of TTX and its analogs in bacteria and shellfish from French coastal waters to contribute to the overall knowledge on levels occurring in shellfish, possible bacterial origin of the toxins and the dynamics of their uptake in situ. LC-MS/MS methodology was optimized for TTX and 5 analogs in bacterial and oyster (*C. gigas*) matrices, paying attention in particular to HILIC-MS/MS performance (i.e., recovery, matrix effect, sensitivity), and verification of stability of TTX in acidic aqueous extracts and oyster matrix. Subsequently, this method was utilized to (i) screen bivalves (oysters *C. gigas* and clams *Ruditapes philippinarum*) collected in 2018 and 2019 at six sites on French mainland coasts and to (ii) screen 122 strains of marine bacteria held in culture collections as a possible source of TTXs. Study sites were selected according to a risk profile as suggested by Turner et al. [30]. Bacterial strains were selected from three separate culture collections at Ifremer, thus drawing on bacteria isolated from either environmental matrices or from shellfish, which were either collected randomly or during shellfish mortality events.

## 2. Results

### 2.1. Method Optimization and Characterization

#### 2.1.1. Analytical Column and Ammonium Formate Concentration

With the conditions described in Appendix A (analytical column, mobile phases and gradient), the S/N ratio of TTX and its analogs (4-epiTTX, 4.9-anhydroTTX and 11-deoxyTTX) was strikingly better with the ZIC-HILIC, followed by the HILIC-Z and BEH-amide columns. Globally, the S/N ratios with the ZIC-HILIC stationary phase for the four TTXs were 7.4–29 times higher when compared to the use of BEH-amide and 3–16 times higher when compared to the use of HILIC Z. Therefore, the ZIC-HILIC column was selected for the remainder of the study. In an attempt to further optimize the signal, we tested three ammonium formate concentrations (5, 10 and 20 mM) in the aqueous mobile phase, and 10 mM gave the best sensitivity. An example chromatogram is provided in Figure 1.

#### 2.1.2. Extraction (Solvent and Volume) and Purification

Overall, the mixture MeOH/water (80:20, *v/v*) gave lower recoveries than acidified water, while recoveries with 1% acetic acid were higher than with 0.2% formic acid (Figure 2). The best recovery for TTX in oyster was obtained with 1% acetic acid (90 ± 24%). Recovery for 4-epiTTX was slightly better in 0.2% formic acid (70 vs. 59%) while MeOH/water/acetic acid (80:20:1, *v/v/v*) was best for 4,9-anhydroTTX (59 ± 3%). Similar recoveries (36–42%) were noted for 11-deoxyTTX, independently of the solvent. As we focused primarily on TTX, 1% acetic acid was chosen as the solvent of extraction, for both shellfish and bacterial matrices.

The volume of extraction with 1% acetic acid was optimized (Figure 3). For oyster, a single extraction with 500 µL was the best compromise when considering both recoveries and repeatability for the four TTXs (77–100 ± 12–15%). Regarding bacteria, better recoveries were observed with a two-step extraction (mean of 97 vs. 79%), however, standard deviations were higher. Therefore, we chose to extract bacteria using a single extraction with 500 µL, as recoveries and repeatability were more homogeneous and to facilitate the screening of the 122 strains.

Concerning the use of an SPE clean-up step, we applied the protocol of Turner et al. [21] but we obtained an abnormally poor recovery for TTX (<5%, *n* = 4) when using the Envi-carb (Supelco) cartridge. Contrastingly, a satisfactory recovery of 65 ± 11% (*n* = 4) was obtained with the GCB cartridge (Phenomenex). Overall, as recoveries of TTX without SPE were better, this desalting and purification step was not implemented here.

Preliminary tests during the optimization of the volume of extraction revealed the presence of a white precipitate with the oyster matrix, especially when performing a two-step extraction. This phenomenon increased when diluting the sample in acetonitrile before LC-MS/MS analysis, as it is classically recommended for HILIC (e.g., Turner et al. [21]). Therefore, an ultrafiltration with a 3 kDa cut-off was used before analysis of oyster samples, while bacteria were ultra-filtered onto 0.22 µm. In addition, as peak shape and retention time were satisfactory without dilution in acetonitrile, samples were injected directly after ultra-filtration (i.e., in acetic acid 1%).

#### 2.1.3. Method Performance

Finally, the optimized method was characterized (Table 1). For the oyster matrix, good to excellent recoveries were obtained (from 77 to 100%), especially for TTX. The matrix effect after filtration onto 3 kDa was toxin-dependent. While a 41% enhancement of the signal was observed for TTX (i.e., by comparing the matrix spiked and the solvent spiked), a moderate to strong ion suppression was noted for the three other analogs (from 29 to 43%). Slightly lower but still satisfactory recoveries were obtained with the bacterial matrix. While all analogs showed ion enhancement after filtration (0.22 µm), the 3.5-fold increase in signal for TTX was abnormally high but very repeatable. Due to the large matrix effects observed, subsequent analysis of field samples was carried out using matrix-matched standards.

With these methods, the LOD and LOQ ranged between 3.8–6.3 and 5–12.5 µg/kg in oyster, with sensitivity being assessed before analyzing each time-series of natural samples, and were 5 and 10 µg/kg in bacterial matrix.

### 2.2. Stability of TTXs in Acetic Acid Extracts and Oyster Tissues

Stability of TTX and the three analogs was generally satisfactory at −20 or 4 °C, since the deviations from the average at −80 °C were within the 95% confidence interval (2 σ) of the value at −80 °C at those temperatures, while significant deviations from the average of storage at −80 °C were observed at 40 °C (Figure 4 and Appendix A).

In particular, TTX suffers from significant degradation in acidic extracts and degradation or irreversible binding in oyster tissues at 40 °C. In acidic extracts, TTX degraded by about 25% over 4 weeks at 40 °C, while degradation in oyster tissue was higher (65% after 4 weeks) and more rapid (approximately 50% degradation in <1 week). On the contrary, 4-epi-TTX concentrations increased significantly at 40 °C, both in acidic extracts (2-fold increase) and in oyster tissues (20-fold increase). The stronger formation of 4-epiTTX in oyster tissue and the stronger degradation of TTX in oyster tissues are coherent with the transformation from one into the other, in particular when considering that the TTX concentration was approximately 100-fold higher than the 4-epiTTX concentration in the standard used for spiking. Formation of 4,9-anhydroTTX (70% increase) was only significant at 40 °C and only in the acidic oyster extract (Appendix A). Degradation of 11-deoxyTTX was outside the 95% confidence interval only in oyster tissue and at 40 °C after 4 weeks (Appendix A). Storage of the acidic extracts was performed both in glass and in polypropylene vials and no difference was found between the two containers for all four analogs (data not shown).

### 2.3. TTXs in Bivalves Collected In Situ

In 2018, TTX was detected in 12 of the 35 samples of bivalves that were screened, 5 time series of oysters and 7 of clams. The concentrations ranged from the LOD to a maximum of 12.2 µg/kg, however, only two oyster samples had a concentration above the LOQ for the three replicates analyzed. These oysters were harvested on 2 July 2018 in the Rance estuary (Brittany, 9.5 ± 2.4 µg/kg) and on 16 July 2018 in the Bay des Veys (Normandy, 8.6 ± 2.5 µg/kg). Interestingly, in the Rance estuary, TTX was detected in at least one oyster replicate between 2 and 17 July (*n* = 3), and between 16 and 30 July in the Bay des Veys (*n* = 2). Noteworthy, no TTX could be detected in Antifer (Bay of Seine, Normandy) while the oysters were sourced from the Bay des Veys. In addition, we also observed low concentrations of TTX (from LOD to 8.3 µg/kg) in all but two samples collected between 18 June and 6 August (*n* = 7, only 1/3 replicate) in the clams sampled from the Rance estuary.

In 2019, TTX was detected in 7 of the 63 bivalves, 6 time-series of oysters and 1 of clams. The concentrations ranged from <LOQ (*n* = 2, one clam and one oyster from the Rance estuary) to 13–32 µg/kg in the other oyster samples (3/3 replicates).

As in 2018, TTX was detected in two (in the Bay of Veys, from 3 to 16 July) or three (in the bay of Brest, from 12 to 21 June) consecutive sampling dates, covering 13 and 9 days, respectively (Figure 5). In all field samples, only TTX was detected. The summary of all our results is provided in Appendix A.

### 2.4. TTXs in Marine Bacterial Strains

By using the method optimized in this study, neither TTX nor C9-based TTX was detected in the 122 marine bacterial strains tested (78% from the genus *Vibrio*). Surprisingly, we also did not detect any TTX in *Vibrio alginolyticus* strain ATCC 17749, even though this strain was reported as a producer of 4,9-anhydroTTX and TTX in Simidu et al. [31] and Pratheepa et al. [32], respectively.

## 3. Discussion

We intended to survey TTX in both shellfish and bacteria from French coastal environments and first devoted significant effort to optimization and characterization of our analytical method for these two matrices, the latter being poorly documented in the literature.

Many of the published methods about the analysis of TTXs in bivalves by HILIC-MS/MS are based on the valuable work of Turner et al. [21] that led to the protocol adopted by the European Union Reference Laboratory for Marine Biotoxins [33] (e.g., [26,28,34,35,36]). This method of reference uses an extraction in acetic acid 1%, a boiling step followed by a graphitized carbon SPE clean-up and a dilution in ACN before analysis, yielding recoveries between 66 ± 2% [37] and 84 ± 18% [21] for the oyster whole flesh matrix and 68 ± 11/20% up to 77 ± 11% for mussel matrix [21,35,38]. In the present study, similar recoveries were obtained after a graphitized carbon SPE step only when using the GCB cartridge (65 ± 11%), while they were unexpectedly poor with the ENVI-carb cartridge despite an identical protocol (<5%). On the contrary, Turner et al. [21] noted that extremely poor recoveries (<15%) were obtained without SPE. However, when considering our results (i.e., 77–100% in oysters and 73–86% in bacteria) and previous reports, the SPE step does not appear to be mandatory to obtain satisfactory recoveries for TTXs (e.g., >90% in bivalve matrices for both Hort et al. [39] and Vlamis et al. [10]). While we also successfully suppressed the boiling step and the dilution before injection, thus simplifying the extraction protocol, we still had to remove proteins via ultrafiltration, especially from the oyster matrix.

Recoveries can significantly be affected by matrix effects. Unfortunately, they were not always assessed in previous studies. Despite the use of an SPE clean-up, the reported matrix effects can still be as high as 61 or 82% for TTX analogs in bivalves (e.g., [38,40]), while a significant dilution of the extract before analysis may be a good alternative (e.g., matrix effects of <9% in mussel and oyster matrices after a 10-fold dilution [39] compared to the 70–80% ion suppression after a two-fold dilution only [27]). These observations confirmed that our sample preparation led to satisfactory matrix effects (ca. a maximum of 40%), except for TTX in the bacterial matrix which showed a repeatable and marked signal increase. This latter condition was still considered appropriate for our screening purpose but will deserve more optimization in the future. Overall, matrix effects for TTXs appear to depend on the analog (−55 vs. +82% for TTX and 11-deoxyTTX in Patria et al. [38], +41 for TTX vs. −29/43% for the other three analogs in oyster here), the matrix (from −45 to +23% in the different organs of oyster [37], −61% in scallop but +29% in mussel in Rey et al. [40], +355% in bacteria here), the analyte concentration (−1 up to +183% at high vs. low concentration in Bordin et al. [35]) and is possibly also chromatography/mass spectrometry-dependent (e.g., −17% in Turner et al. [21] vs. −55% in Patria et al. [38], based on comparison of published data with the *EURLMB*-protocol). Therefore, matrix effects should be determined and corrected, especially if matrix-matched calibration curves are not used.

Generally, the limits of detection and quantification reported for HILIC-MS/MS methods by other groups were in the low µg/kg range (Appendix A), similar to the sensitivity of our method (i.e., 3–12.5 µg/kg) and below the EFSA recommended maximum permitted level of 44 µg TTX-eq./kg [11]. It should be noted that in our case, the use of the ZIC-HILIC (Merck) chromatographic column allowed us to lower the limits of detection and quantification, especially compared to the BEH Amide column widely used so far. Still, several groups achieved an excellent sensitivity <1 µg/kg (e.g., [21,28,41]), probably as a result of a better performance of their mass spectrometers as both recovery and matrix effects were satisfactory here. Indeed, the combination of good recovery, appropriate matrix effects and sensitive enough mass spectrometer is critical to obtain adequate LC-MS/MS methods for the quantification of TTXs in bivalves and bacteria.

Then, we used our fit-for-purpose optimized method to analyze the bivalves and bacteria collected or isolated in French shellfish growing areas. In fact, TTX has already been reported in mussels sampled in France in 2018 (<11 µg/kg, *n* = 3/103 positive samples), but not in oysters (*n* = 0/24) [39]. While we confirmed the episodic presence of TTX in French bivalves, we reported that oysters (*C. gigas*, *n* = 5/26 and 6/50 in 2018 and 2019) and clams (*R. philippinarum*, *n* = 7/8 and 1/13 in 2018 and 2019) can also contain TTX, at relatively, but similar, low concentrations (i.e., from <LOQ to 32 µg/kg) and in other sites compared to the previous French study. Our results are also in agreement with the occurrence and concentration of TTXs at a larger scale (Appendix A). Indeed, although thousands of bivalves have been screened for TTX(s) in Europe and elsewhere, only a limited number of samples was found positive and in a limited number of locations, with generally low concentrations (i.e., dozens of µg/kg, e.g., [28,39,42,43]), although there are a few exceptions. When considering only results obtained by LC-MS, the highest TTX concentration of 870 µg/kg was observed in the clam *Paphies australis* in New Zealand [44], followed by 541 µg/kg (i.e., 413 µg/kg by LC-HRMS) in one sample of the mussel *Mytilus galloprovincialis* from the Northern Adriatic Basin, Italy [35] while 253 µg/kg were observed in the Pacific oyster *Crassostrea gigas* from both the South of England [30] and the South-West of the Netherlands [45]. However, there are some unusual reports about the presence and duration of toxic episodes. Notably, the clam *P. australis* seems to harbor TTX from several locations in New Zealand [46], including at different periods of the year at the hotspot of Hokianga Harbor [42]. Similarly, TTX was always detected in the digestive gland of the Japanese scallop *Patinopecten yessoensis* over an 8-month period of weekly monitoring in the Bay of East Japan [47]. Finally, at a smaller timescale, a shellfish production area with a higher prevalence of TTX was identified in Southern England (i.e., presence in oysters for more than 9 weeks in a row in 2019) [30,37]. Although there may be site-specific conditions favoring the presence of TTXs in these areas, the observations from Biessy and colleagues [36,46], Numano et al. [47] and Dhanji-Rapkova et al. [37] point towards a food source origin of TTX in clams, scallops and oysters, respectively.

Among the potential primary source(s) of TTXs, the bacterial origin seems more widely supported than the dinoflagellate one, as more than 150 bacterial strains have been reported to produce TTX [24] while there is no direct evidence of a dinoflagellate production. As a result, we screened 122 strains of bacteria covering 39 genera and 99 species, including main genera of potential TTX-producers (i.e., *Vibrio*, *Bacillus* and *Pseudomonas*) [24]. However, no TTX could be detected in our conditions. We focused primarily on bacteria of the genus *Vibrio* as 9/10 strains of *V. parahaemolyticus* and 1/1 of *V. cholerae* were successfully isolated from oysters and mussels and proved to produce TTX in culture [9]. While we did not isolate our bacteria from TTX-containing bivalves, we still analyzed 84 strains of *Vibrio*, including from the abovementioned species and also from the *V. alginolyticus* species which was recently reported as a TTX-producer despite being isolated from a TTX-negative worm [48]. Moreover, our bacterial strains came from many different places and environments along the French coasts (Appendix A) and included environmental, pathogenic or ‘constitutive’ representatives as the exact origin (i.e., endogenous or exogenous) of TTX is elusive. The observation that none of the selected bacteria produced TTX, similarly to 0/106 strains (58% of *Vibrio*) in Pratheepa et al. [32], raises several questions.

(1)Is the correlation between the presence of *Vibrio* spp. and TTX relevant and the presence of PKS/NRPS genes informative? These links have been tested but contradictory results were reported [9,26,27,28,32].(2)Are we overestimating the number of bacterial TTX-producing strains? Some of the bacteria mentioned in the review of Magarlamov et al. [24] might have been incorrectly reported as a TTX-producer, as for example a strain of *V. alginolyticus* that was classified as a false-positive due to the presence of a low molecular weight compound with the same molecular weight as TTX [49]. Similarly, we were not able to detect any TTXs in the *V. alginolyticus* strain ATCC 17749 although considered as a producer of 4,9-anhydroTTX [31] and TTX [32]. We did use culture conditions identical to those of the latter group, however, chromatographic evidence may need to be carefully reviewed before concluding on the toxin production of this strain. This is why highly specific and reliable methods are recommended to ascertain the presence of TTXs. We assumed our method was adapted for our screening purpose, both in terms of recovery and sensitivity, although showing a high signal enhancement due to the bacterial matrix.(3)Are we using the appropriate culture conditions to elicit TTX production? Indeed, Chau et al. [23] stated that bacteria could require specific inducers to promote TTX production. As discussed by Melnikova et al. [50], the medium content including phosphate concentration, temperature, growth phase and duration of cultivation may modulate the TTX content of bacteria. Turner et al. [48] confirmed that lower temperature (i.e., 22 °C compared to 23.5 and 41 °C) were required to detect TTX in *P. luteola* and *V. alginolyticus* isolated from worms. The production and accumulation of TTX-like compounds (revealed by confocal laser scanning microscopy using anti-TTX-antibodies) in *Bacillus* sp. 1839 was restricted to spores and sporulation would ultimately result in the presence of TTX in the ribbon worm *Cephalothrix simula* [51]. This *Bacillus* sp. strain appears to continuously produce TTX over time, as confirmed by the presence of TTX in spore-enriched cultures by LC-MS/MS, which makes this specific strain unique according to the authors [50]. Indeed, some studies revealed the complete loss of toxin production when bacteria were cultivated on artificial media (see references in Magarlamov et al. [24]). Finally, the possibility that unculturable microorganisms play a role in TTX biosynthesis cannot be ignored [23].(4)Should we focus on other source(s) of TTXs? It was suggested that some dinoflagellates, namely *Alexandrium tamarense* [52] and *Prorocentrum cordatum* (formerly *P. minimum* [10]) could be other sources of TTX, as they were concomitantly present at significant concentrations at the time when TTX-containing bivalves were harvested. Again, this hypothesis of a link between either *P. cordatum* or *A. tamarense* was tested but not confirmed or even refuted [30,35,39,47]. The search for the origin of TTX in the Greek mussels led to the discovery of two C9-based TTX-like compounds in cultivated strains of *P. minimum* thanks to Precursor Ion scan analyses [29]. The authors suggested that the origin was “associated with bacteria”. Unfortunately, we could not detect any C9-based TTX in our bacteria. Recently, picocyanobacteria (especially the genera *Synechococcus*, *Cyanobium* and *Prochlorococcus*) were proposed as a new hypothetic source of TTX, after they were detected in 70–90% of the core bacterial communities of TTX-bearing *P. australis* in New Zealand, but not in the sympatric and non-TTX containing clam *Austrovenus stutchburyi* [53]. The hypothesis of an algal source of TTX is also potentially confounded with the increasing knowledge on bacterial components of the phycosphere [54,55,56,57], which may result in microalgal transport of bacterial toxins into bivalves. In addition, as bivalves are not efficient at filtering free-living bacteria directly from seawater as a food source due to their small size [58], it is reasonable to assume that if bacteria are the biological source they may enter the bivalve digestive system as “back-packers”, i.e., as part of the phycosphere or from protists [59] or aggregates [60] or via another vector such as toxic flatworm larvae as recently proven by Okabe et al. [61].

In summary, further work will be required to understand the relationship between bivalves, bacteria and TTX. A special focus should be put on the biological source(s) of TTX and particularly the obtention of TTX-producing bacteria or other microorganisms. Based on the valuable studies of Turner and colleagues, isolation of bacteria from both bivalves and adjacent sediments should be attempted and screened for TTXs. Attention should not be only paid to the genus *Vibrio* since other genera also have TTX-producing representatives (e.g., *Aeromonas*, *Shewanella*, *Pseudomonas* or *Alteromonas* [62]). However, regarding the relatively low number of bivalves containing TTXs (Appendix A), the low number of TTX-producing bacterial strains compared to all the bacteria that were screened for, and considering that bacteria in isolated cultures may lose the ability to produce TTX over time [63] this might be a tedious and time-consuming approach. Still, the efforts should target some ‘hotspots’ where TTX is regularly detected, such as in Southern England [37] and more significantly in Hokianga Harbor, New Zealand [42,46]. In addition, any factor triggering a bacterial stress response should be better explored. The comparison of bacterial communities between TTX-bearing and non-TTX-bearing organisms [48,53,64] can pinpoint potential TTX-producing species. Nevertheless, the metabarcoding approach may not provide the appropriate level of identification as TTX production does not seem to be a common feature among a given species but more strain-specific, and TTX-producing bacteria can still be isolated from non-TTX-bearing organisms [48]. The discovery of the TTX biosynthetic pathway, and the genes associated with this pathway would certainly provide a more definitive possibility to screen for TTX-producing bacteria. Hopefully, the availability of *Bacillus* sp. 1839 (now referred to as *Cytobacillus gottheilii*) genome [65] will facilitate this process.

So far, TTXs were only occasionally detected in bivalves of European waters (Appendix A). Despite many studies in different EU countries, the number of samples exceeding the EFSA recommendation is very limited since the first report of TTXs in bivalves in the United Kingdom and Greece in 2015 [9,10], which may reflect an overall low risk for the consumers. However, occasionally, significant concentrations (>250 µg/kg) have been detected in shellfish from UK, Dutch and Italian growing areas [30,35,45], and if these co-occurred with other paralytical shellfish toxins, e.g., of the STX-group, the combined toxic equivalent would be relevant to public health. In fact, the concomitant presence of these toxins has already been reported e.g., in Southern Europe and Japan [43,47].

Based on the other screening studies published and our results, it seems that short-term episodes (generally less than 4 weeks) at the beginning of summer are predominant in “shallow, estuarine waters with temperatures above 15 °C” [30,45]. However, as systematic monitoring is not being conducted and very few data exist prior to 2015, it appears premature to conclude that “TTX levels in mollusks appear to have decreased in general” since 2017 as stated by Hort et al. [39]. Interestingly, here TTX was detected both years in the Rance estuary (Brittany) and the Bay of Veys (Normandy), and in 2019 in the Bay of Brest (Brittany). Consequently, the Bay of Veys will be included in the French national surveillance program EMERGTOX. In addition, the monitoring of the Bay of Brest may be pursued in additional studies, to confirm the presence of TTX over time (i.e., where the present study found the highest concentration of 32 µg TTX/kg) and because there is an emerging risk of *A. minutum* blooms and shellfish contamination with STXs [66].

Based on our observations, we could recommend that the short duration of the contamination events makes frequent sampling necessary, i.e., weekly sampling should be adopted as a minimum in further studies. In addition, our stability study showed that heat-induced transformation of TTX into 4-epiTTX was more pronounced in shellfish tissues than in acidic solution, and therefore, any survey should ensure that shellfish samples are shipped frozen on ice, with temperatures on arrival not exceeding 4 °C.

## 4. Materials and Methods

### 4.1. Samples

#### 4.1.1. Sampling of Bivalves

Sites for sampling (Figure 6) were selected by crossing the environmental parameters available in the REPHY (Observation and Surveillance Network for Phytoplankton and Hydrology in coastal waters, Ifremer) database with the conclusions of Turner et al. [30] who observed “a greater level of risk in areas of shallow, estuarine waters with temperatures above 15 °C”.

While all selected sites are coastal or estuarine and water temperatures typically exceed 15 °C in late spring and early summer, salinity fluctuations were generally smaller than those reported by Turner et al. [30].

Oysters (*Crassostrea gigas*) and clams (*Ruditapes philippinarum*) were sampled over two consecutive years in 2018 and 2019 (Table 2).

Samples corresponded to at least 10 and 30 individuals for oysters and clams, respectively. They were sent frozen to the Phycotoxins laboratory, homogenized and stored at −20 °C until further processing.

#### 4.1.2. Bacteria from Culture Collections

In total, 122 strains were cultivated in three Ifremer laboratories holding bacterial collections (LSEM, LGPMM and IHPE), taxonomically covering 39 genera and 99 species (Appendix A). They were classified as ‘environmental’ (isolated from the water column or sediment), ‘pathogenic’ (isolated from 6 mollusk species or seawater during mortality episodes) or ‘constitutive of the microbiome’ (isolated from the hemolymph of healthy oysters).

In an attempt to select the culture medium, we first compared the effect of two media used to retrieve marine heterotrophic bacteria (i.e., Marine Broth (Difco 2216, Fisher Scientific, Illkirch, France) and Zobell [67]) on the production of TTXs by *Vibrio alginolyticus* strain ATCC 17749 (reported as a 4,9-anhydroTTX and TTX producer in Simidu et al. [31] and Pratheepa et al. [32], respectively). Unfortunately, no toxin was detected but this strain was used as the bacteria blank matrix. The Zobell medium was chosen as it was less expensive and more convenient to handle.

Overnight cultures of 100 mL were performed at 20, 22 or 37 °C in Zobell medium then the cells were centrifuged 15 min at 5000× g and the bacterial pellets stored at −20 °C until toxin extraction.

### 4.2. Chemicals and Reagents

Acetic acid (AA, >99% purity) and ammonium formate (AF, 10 M solution) were purchased from Sigma-Aldrich (Saint-Quentin-Fallavier, France) and formic acid (FA, >98% purity) from Fluka (Steinheim, Germany). Acetonitrile and methanol were of LC-MS grade (Honeywell, Seelze, Germany) and milliQ water was supplied by a Milli-Q integral 3 system (Millipore, Saint-Quentin-en-Yvelines, France).

The TTX standard solution (CRM-03-TTXs) was from CIFGA (Lugo, Spain) and contained 25.1 ± 1.2 µg/mL TTX, together with 2.98 ± 0.16 µg/mL 4,9-anhydroTTX, 0.76 µg/mL 11-deoxyTTX, 0.75 µg/mL 4-epi-TTX and trace levels of 11-norTTX-6-ol (norTTX) and 5,6,11-trideoxyTTX (trideoxyTTX).

### 4.3. Method Optimization

#### 4.3.1. Solvent and Volume of Extraction

Four solvents or mixtures (500 µL of acetic acid 1% in water or in MeOH 80%, formic acid 0.2% in water or in MeOH 80%) were compared by spiking the oyster matrix while the volume (500, 700 or 900 µL in 1 or 2 extraction steps) was optimized using both matrices (oyster and bacteria). Matrices were spiked in duplicate and injected twice. The final concentration was 500, 60 and 15 ng/mL for the optimization of the solvent, and 60, 7.5 and 1.9 ng/mL for the optimization of the volume, for TTX, 4,9-anhydroTTX and 4-epiTTX/11-deoxyTTX, respectively.

#### 4.3.2. Purification

We tested the recovery with two different SPE cartridges: ENVI-carb (Supelco, Sigma-Aldrich, Saint-Quentin-Fallavier, France) and GCB (Phenomenex, Le Pecq, France) following the protocol of Turner et al. [21] (except that the sample loaded was 1 mL). In parallel, we also tested filtration (3 kDa or 0.2 µm, Nanosep or Nanosep MF, Pall, Saint-Germain-En-Laye, France) instead of SPE.

#### 4.3.3. Analytical Column and Mobile Phases

Three HILIC columns were compared (Appendix A): BEH-amide (150 × 2.1 mm, 1.7 µm, Acquity UPLC, Waters Guyancourt, France), HILIC-Z (100 × 2.1 mm, 1.7 µm, Poroshell 120, Agilent Technologies, Les Ulis, France) and ZIC-HILIC (150 × 2.1 mm, 3.5 µm, SeQuant^®®^, Merck, Fontenay-sous-Bois, France). The BEH-amide column was tested according to the mobile phases and gradient of Turner et al. [21]. The two other columns were tested as mentioned above, except that the gradient for the HILIC-Z was slightly modified (starting with 70% of A) and the ammonium formate concentration was 20 mM. To select the analytical column giving the best signal, a mix of standards (concentrations of 126 ng/mL for TTX, 15 ng/mL for 4,9-anhydroTTX and 3.8 ng/mL for 4-epiTTX and 11-deoxyTTX) was injected and the signal-to-noise S/N of the quantitative transition was compared. Finally, the effect of the concentration of ammonium formate (5, 10, 20 mM) in the mobile phase A was tested on the retention of TTXs and intensity of the chromatographic peaks.

### 4.4. Optimized Extraction and Analysis

#### 4.4.1. Extraction of TTXs from Shellfish and Bacteria

For the extraction of TTXs, glass beads (250 mg, 0.15–0.25 mm, VWR) and 1% aqueous acetic acid (0.5 mL) were added to 200 mg of shellfish (oyster or clam) or 50 mg of bacterial cell pellet. The sample was briefly vortexed and ground using a mixer mill (MM400, Retsch, Eragny, France) for 10 min at 30 Hz. Tubes were centrifuged (15,000× *g*, 10 min) and debris-free supernatants were ultra-filtered onto 3 kDa (Nanosep, Pall) for shellfish and 0.22 µm (Nanosep MF, Pall) for bacteria.

#### 4.4.2. HILIC-MS/MS Analysis of TTXs

TTX analysis was performed by ultrafast liquid chromatography (Nexera, Shimadzu), coupled to a hybrid triple quadrupole/linear ion-trap mass spectrometer (QTRAP 5500, Sciex, Villebon-sur-Yvette, France) equipped with a TurboV electrospray ionization source.

The chromatographic column was a ZIC^®®^-HILIC (150 × 2.1 mm, 3.5 μm, SeQuant^®®^, Merck) equipped with a guard column (ZIC^®®^-HILIC, 50 × 2 mm, SeQuant^®®^, Merck). The column and sample temperatures were 40 and 4 °C, respectively. The flow rate was 0.3 mL/min, and the volume of injection was 5 μL. The binary gradient consisted of (A) water containing 0.1% formic acid and 10 mM ammonium formate and (B) acetonitrile containing 0.1% formic acid (*v/v*). The elution gradient started with 40% A (0–1 min), increased from 40 to 50% A (1–5 min), maintained for 2 min (5–7 min) and decreased at 40% A for subsequent re-equilibration (7–10 min).

The MS/MS detection was performed in positive ionization mode using multiple reaction monitoring (MRM). The source parameters were curtain gas 30 psi; temperature 650 °C; Gas 1 40 psi; Gas 2 60 psi; ion spray voltage 5500 V. For detection, the parameters are provided in Table 3. Samples were quantified using matrix-matched calibration curves prepared in oyster or bacteria blank matrix (spiked before the extraction), with concentrations ranging between 2.5 and 250 ng/mL for TTX, 2.4–30 ng/mL for 4,9-anhydroTTX and 0.60–7.6 ng/mL for both 4-epiTTX and 11-deoxyTTX. These concentrations corresponded to 6.3–627, 6–74.5and 7.5–18.8 µg/kg for TTX, 4,9-anhydroTTX and 4-epiTTX/11-deoxyTTX in shellfish and 2.5–251, 2.4–29.8 and 0.60–7.6 µg/kg for TTX, 4,9-anhydroTTX and 4-epiTTX/11-deoxyTTX in bacteria. The identity of each TTX analogue was confirmed based on the retention time, the two mass spectral transitions and the ion ratio between the quantitative and qualitative transitions, with a 20% tolerance from the value of reference.

All samples were first screened, and when TTX was detected, two other replicates were extracted and analyzed.

Acquisition and data processing were performed using Analyst 1.6.3 software (Sciex Villebon-sur-Yvette, France).

### 4.5. Characterization of the HILIC-MS/MS Method

#### 4.5.1. Limit of Detection and Limit of Quantification

Limit of detection (LOD) was defined as the lowest concentration giving a signal-to-noise (S/N) ratio of 3 for the qualitative mass spectral transition. LOD was assessed only for TTX (i.e., *m/z* 320 > 162) at the time of the analysis (i.e., in 2019 and 2020).

The limit of quantification (LOQ) was defined as the lowest concentration giving a S/N of 10 for the quantitative mass spectral transition. LOQ was assessed only for TTX (i.e., *m/z* 320 > 302).

#### 4.5.2. Recoveries and Matrix Effects

Recovery was defined as the ratio of areas obtained between a blank matrix spiked before extraction and an extract of a blank matrix (i.e., oyster or bacteria) spiked before analysis. Matrix effect was the ratio of areas between an extract spiked before analysis and the solvent (acetic acid 1%) spiked at the same concentration (the final concentrations used were 63, 7.5 and 1.9 ng/mL for the 4 analogues, i.e., 158, 18.8 and 4.8 µg/kg of shellfish and 63, 7.5 and 1.9 µg/kg of bacteria).

### 4.6. Stability of TTXs in Acetic Acid Extracts and Oyster Tissues

In order to assess possible storage phenomena of tissues or extracts in a routine analysis setting, stability of both crude extracts and homogenized oyster tissues was evaluated in an accelerated stability study over a 4-week period. Extracts in glass versus polypropylene (PP) vials and fresh homogenate of oyster in glass vials were spiked with TTXs (concentrations of 250, 30 and 7.5 ng/mL for TTX, 4,9-anhydroTTX and 4-epiTTX/11-deoxyTTX) and stored at 3 temperatures (−20, 4 and 40 °C) over 4 weeks to test the stability of the analytes in these conditions. A reverse isochronous design (as reported by Lamberty et al. [68]) was used. Each week, three replicates of each condition were placed at −80 °C, and a triplicate was also maintained at −80 °C at the start of the experiment and used as the reference. After the 4-week period, the fresh homogenate was extracted as abovementioned and all the samples were analyzed using the optimized HILIC-MS/MS method. In order for the stability studies to be performed under repeatability conditions, the analysis of all time points was carried out in a single analytical batch. Results were expressed as the ratio of the mean of areas (*n* = 3) for each condition assessed compared to the mean of the triplicate maintained at −80 °C during the four weeks.

## Figures and Tables

**Figure 1 toxins-13-00740-f001:**
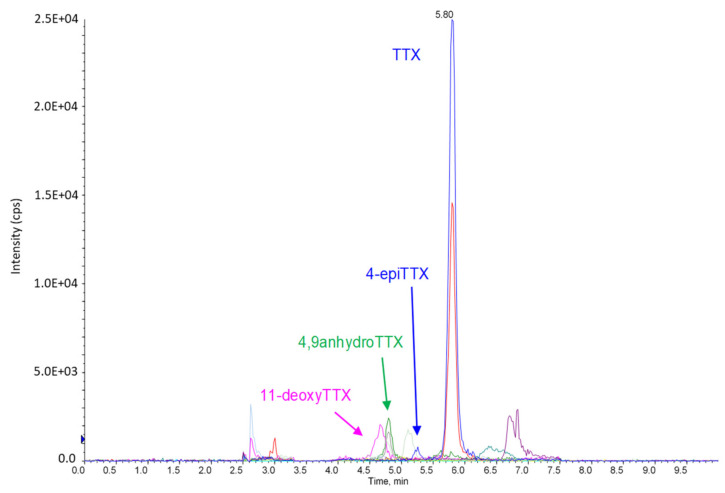
Chromatogram obtained with the optimized conditions for TTXs spiked in the oyster matrix (highest concentrations of the calibration curves).

**Figure 2 toxins-13-00740-f002:**
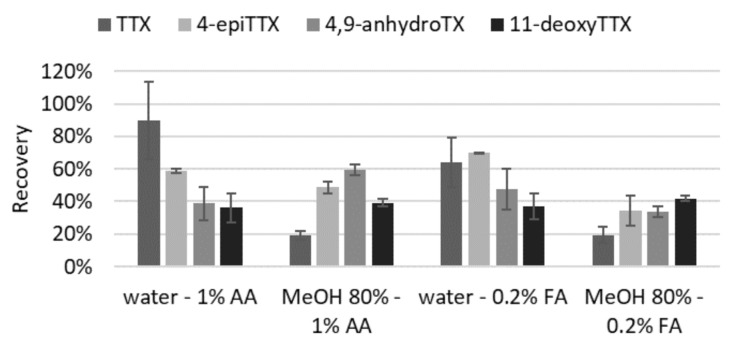
Recovery obtained for TTX and analogs in an oyster blank matrix according to the solvent of extraction (AA: acetic acid, FA: formic acid).

**Figure 3 toxins-13-00740-f003:**
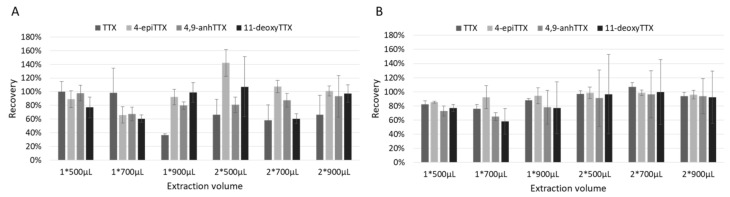
Optimization of the volume of extraction of TTXs for (**A**) oyster and (**B**) bacterial matrices.

**Figure 4 toxins-13-00740-f004:**
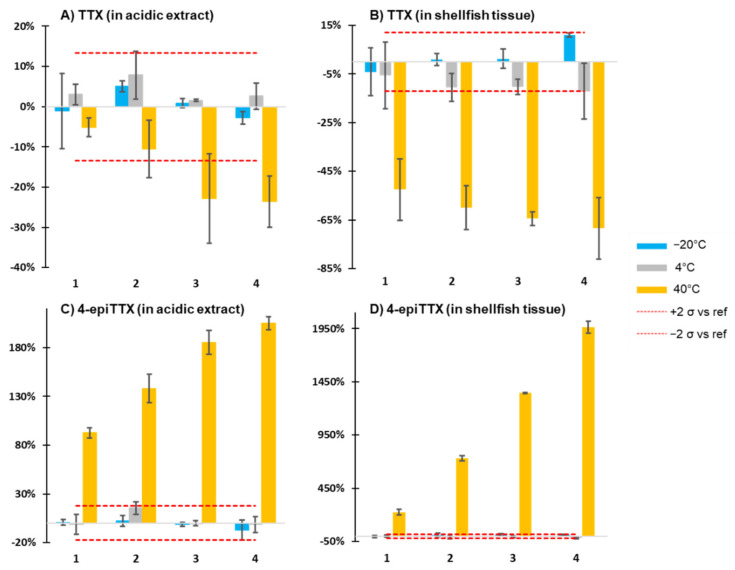
Four-week stability study of TTX and 4-epiTTX spiked into acetic acid extract of blank oyster matrix (stored in glass vials) or blank oyster matrix at −80, −20, 4 and 40 °C. Values represent % deviations from the average of the −80 °C reference condition. (**A**) TTX spiked into acetic acid oyster extract, (**B**) TTX spiked into blank oyster matrix, (**C**) 4-epiTTX spiked into acetic acid oyster extract and (**D**) 4-epiTTX spiked into blank oyster matrix. Error bars represent the standard deviation (*n* = 3), red dotted lines delineate the confidence interval (95%, 2σ) of the −80 °C reference condition (*n* = 12).

**Figure 5 toxins-13-00740-f005:**
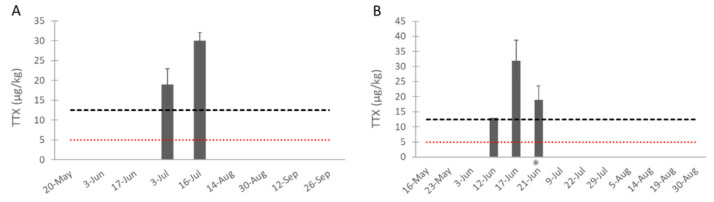
Concentration of TTX (mean ± SD, *n* = 3, except on 12 June in B due to an injection issue) in the oysters sampled in 2019 from (**A**) the Bay of Veys (Normandy, bi-weekly sampling) and (**B**) the Bay of Brest (Brittany, weekly sampling). The LOD (red line) and LOQ (black line) were represented. No bar means the value was <LOD (sampling was carried out for the periods shown). * Note that a second lot of oysters was used from this date, due to an unexplained mortality event.

**Figure 6 toxins-13-00740-f006:**
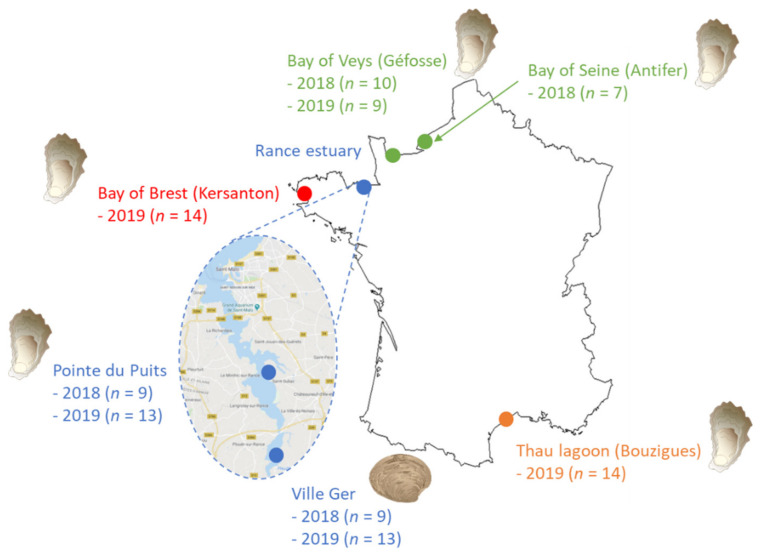
Sites for in situ sampling of oysters and clams in 2018 and 2019 (numbers in brackets indicate the number of consecutive weekly or biweekly sampling occasions).

**Table 1 toxins-13-00740-t001:** Recovery, matrix effect, limit of detection and quantification (LOD/LOQ) of TTXs in oyster and bacterial matrix.

Analog	Oyster			Bacteria		
	Recovery (%)	Matrix Effect (%)	LOD/LOQ (µg/kg)	Recovery (%)	Matrix Effect (%)	LOD/LOQ (µg/kg)
TTX	100 ± 15	141 ± 25	3.8/5 (2019) 6.3/12.5 (2020)	82 ± 5	355 ± 1	5/10
4-epiTTX	89 ± 12	71 ± 20	nd	86 ± 1	123 ± 16	nd
4,9-anhydroTTX	98 ± 12	57 ± 12	nd	73 ± 7	117 ±26	nd
11-deoxyTTX	77 ± 15	58 ± 12	nd	77 ± 5	128 ± 23	nd

nd: not determined.

**Table 2 toxins-13-00740-t002:** Sites for in situ sampling of oysters and clams in 2018 and 2019, including location, bivalve species and number of samples collected each year.

Location	Year	Dates	Bivalve Species	Number of Samples
Antifer * (Bay of Seine)	2018	From 12 July to 9 October (bi-weekly)	Oyster (*C. gigas*)	7
Géfosse (Bay of Veys)	2018 2019	From 13 June to 24 October (bi-weekly) From 20 May to 26 September (bi-weekly)	Oyster (*C. gigas*)	10 9
Pointe du Puits (Rance estuary)	2018 2019	From 11 June to 6 August (weekly) From 3 June to 26 August (weekly)	Oyster (*C. gigas*)	9 13
Ville Ger (Rance estuary)	2018 2019	From 11 June to 6 August (weekly) From 3 June to 26 August (weekly)	Clam (*R. philippinarum*)	9 13
Kersanton ** (Bay of Brest)	2019	From 16 May to 30 August (weekly)	Oyster (*C. gigas*)	14
Bouzigues (Thau lagoon)	2019	From 27 May to 28 August (weekly)	Oyster (*C. gigas*)	14

* oysters were transferred from Géfosse and allowed to acclimate for one month before sampling as this is not a shellfish farming area but it matched the observations of Turner et al. [30]. ** oysters were sourced from the bay of Brest and acclimated for 14 days before starting the sampling. A second batch of oysters was used after an unexplained mortality event and collected after 18 days of acclimation.

**Table 3 toxins-13-00740-t003:** Compound parameters for the analysis of TTXs by HILIC-MS/MS.

Analytes	Q1 Mass (*m/z*)	Q3 Mass (*m/z*)	Transition Type	DP (V)	CE (eV)	CXP (V)	Ion Ratio (Quant/Qual)
TTX and 4-epiTTX	320	302	Quant	86	35	48	TTX = 1.9 4 epiTTX = 3.0
		162	*Qual*	86	59	26	
4,9-anhydroTTX	302	162	Quant	121	47	26	1.3
		256	*Qual*	121	37	44	
11-deoxyTTX	304	286	Quant	86	35	22	6.3
		176	*Qual*	86	47	28	
norTTX	290	272	*Qual*	121	35	22	/
		162	*Qual*	121	35	22	
TrideoxyTTX	272	254	*Qual*	86	47	28	/
		162	*Qual*	86	47	28	
C9-based-TTX (265)	265	179	*Qual*	86	59	26	/
		162	*Qual*	86	59	26	
C9-based-TTX (308)	308	180	*Qual*	86	59	26	/
		162	*Qual*	86	59	26	

DP: declustering potential; CE: Collision energy; CXP: Collision cell eXit Potential Quant: quantitative transition; Qual: qualitative transition.

## Data Availability

REPHY dataset - French Observation and Monitoring program for Phytoplankton and Hydrology in coastal waters. Metropolitan data. SEANOE. https://doi.org/10.17882/47248, accessed on 28 May 2018.

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
