# Peer review of "Tetrodotoxins in French Bivalve Mollusks—Analytical Methodology, Environmental Dynamics and Screening of Bacterial Strain Collections"

_toxins, 2021, doi:10.3390/toxins13110740_

Round 1
Reviewer 1 Report
The authors made an in-depth review of the literature on the basis of which they wrote the introduction to the article. The research was properly planned. The research methods used were well described. The assumed research goals were achieved. The obtained test results have been properly interpreted and documented. The statistical analysis of the results is correct. Based on the obtained results, the authors formulated correct conclusions.
The authors tested the presence of tetrodotoxins (TTXs) in 98 shellfish samples and 122 bacterial strains from French coastal environments. They developed a procedure for the determination of TTXs in these two matrices. They performed a single extraction followed by ultrafiltration and SM analysis. They validated the method by providing the detection and quantification limit of TTXs. They estimated that shellfish samples should be shipped frozen on ice, with temperatures not exceeding 4 ° C when the shipment reaches its final destination.
The quality of the food we eat has a huge impact on our health. There are close relationships between food, nutrition and human health. Our health depends on how we eat and what kind of food we provide to the body. This is an undeniable fact that is not worth arguing with. Properly performed food analysis is very important for humans. First of all, it is responsible for food safety, i.e. all the rules in force regarding its proper production, the purpose of which is to guarantee the health of consumers. In order for food to be considered safe and to be allowed for sale, it must undergo a detailed analysis to assess its quality. In order to be sure that the food is free from harmful chemical compounds and its consumption does not contribute to the development of various types of diseases that have a negative impact on the health of the consumer, it should be subjected to a multistage analysis. For this reason, the research undertaken in the reviewed manuscript has practical application in food analysis. This is the main reason why this work should be published. Test results are very well documented in Supplementary Materials. Which is also a strong advantage of this manuscript, as it indicates the reliability of the research carried out. A considerable novelty of this work is the development of a procedure for determining TTXs in bacterial strains, because so far this matrix has been poorly described in the scientific literature. The authors also made an in-depth discussion of the obtained results based on the available scientific literature. The only small drawback of this work is that the samples were from 2018-2019. However, the lack of samples from 2020 and 2021 does not detract from the value of this manuscript.
Author Response
Reviewer 1
Open Review
(x) I would not like to sign my review report
( ) I would like to sign my review report
English language and style
( ) Extensive editing of English language and style required
( ) Moderate English changes required
( ) English language and style are fine/minor spell check required
(x) I don't feel qualified to judge about the English language and style
|
Yes |
Can be improved |
Must be improved |
Not applicable |
|
|
Does the introduction provide sufficient background and include all relevant references? |
(x) |
( ) |
( ) |
( ) |
|
Is the research design appropriate? |
(x) |
( ) |
( ) |
( ) |
|
Are the methods adequately described? |
(x) |
( ) |
( ) |
( ) |
|
Are the results clearly presented? |
(x) |
( ) |
( ) |
( ) |
|
Are the conclusions supported by the results? |
(x) |
( ) |
( ) |
( ) |
Comments and Suggestions for Authors
The authors made an in-depth review of the literature on the basis of which they wrote the introduction to the article. The research was properly planned. The research methods used were well described. The assumed research goals were achieved. The obtained test results have been properly interpreted and documented. The statistical analysis of the results is correct. Based on the obtained results, the authors formulated correct conclusions.
The authors tested the presence of tetrodotoxins (TTXs) in 98 shellfish samples and 122 bacterial strains from French coastal environments. They developed a procedure for the determination of TTXs in these two matrices. They performed a single extraction followed by ultrafiltration and SM analysis. They validated the method by providing the detection and quantification limit of TTXs. They estimated that shellfish samples should be shipped frozen on ice, with temperatures not exceeding 4 ° C when the shipment reaches its final destination.
The quality of the food we eat has a huge impact on our health. There are close relationships between food, nutrition and human health. Our health depends on how we eat and what kind of food we provide to the body. This is an undeniable fact that is not worth arguing with. Properly performed food analysis is very important for humans. First of all, it is responsible for food safety, i.e. all the rules in force regarding its proper production, the purpose of which is to guarantee the health of consumers. In order for food to be considered safe and to be allowed for sale, it must undergo a detailed analysis to assess its quality. In order to be sure that the food is free from harmful chemical compounds and its consumption does not contribute to the development of various types of diseases that have a negative impact on the health of the consumer, it should be subjected to a multistage analysis. For this reason, the research undertaken in the reviewed manuscript has practical application in food analysis. This is the main reason why this work should be published. Test results are very well documented in Supplementary Materials. Which is also a strong advantage of this manuscript, as it indicates the reliability of the research carried out. A considerable novelty of this work is the development of a procedure for determining TTXs in bacterial strains, because so far this matrix has been poorly described in the scientific literature. The authors also made an in-depth discussion of the obtained results based on the available scientific literature. The only small drawback of this work is that the samples were from 2018-2019. However, the lack of samples from 2020 and 2021 does not detract from the value of this manuscript.
Submission Date
05 October 2021
Date of this review
12 Oct 2021 19:09:31
We would like to thank Reviewer 1 for his appreciation of our work and for considering the broader importance of such method development and associated results concerning food safety issues.
Reviewer 2 Report
The paper aimed to determine the presence of TTXs in French Bivalve Mollusks and bacteria, and to understand its putative sources.
In general, the manuscript is well written and organized. The manuscript has a small number of figures, and I believe some of the figures supplied in the supplementary material (such as Figs. S1 or S3, for example), could be moved to the main text.
It requires additional clarifications of several aspects:
Table 1 – regarding the amount of toxin reported in bacterial strains, I question the presentation of results: were pellets weighed to allow the expression of results in µg/kg, as in bivalves? No details were given in section 5.1.2 as to how the number of bacterial cells in each culture was counted (or weighed).
Line 221 – I suggest removing the comment ‘unfortunately’. ‘Luckily’, coastal waters are not full of toxin-producing microorganisms, and as result, we can get some food from coastal waters (such as bivalves)!
Section 4- I enjoyed the detailed discussion on the null results obtained. I wonder the ‘Arcachon’ mystery (of positive mouse bioassays), that lasted for a few years, was not related to TTXs after all? The authors could comment on this episode here, as, on some occasions, explanations for certain phenomena are not easy to get.
Section 5.1.2 / Table 4 - explain the abbreviations used, please.
Line 454 – the supplier for the Zobell medium was not detailed.
Section 5.3.2 – the writing is confusing. The SPE procedure was used, but also not used (filtration was used in alternative). All of these experimental steps were mixed in a single sentence. Please clarify.
Section 5.3.3. – contains the description of the LC columns used. This was repeated in section 5.4.2. Please merge sections.
Author Response
Reviewer 2
Open Review
( ) I would not like to sign my review report
(x) I would like to sign my review report
English language and style
( ) Extensive editing of English language and style required
( ) Moderate English changes required
(x) English language and style are fine/minor spell check required
( ) I don't feel qualified to judge about the English language and style
|
Yes |
Can be improved |
Must be improved |
Not applicable |
|
|
Does the introduction provide sufficient background and include all relevant references? |
(x) |
( ) |
( ) |
( ) |
|
Is the research design appropriate? |
(x) |
( ) |
( ) |
( ) |
|
Are the methods adequately described? |
( ) |
( ) |
(x) |
( ) |
|
Are the results clearly presented? |
( ) |
(x) |
( ) |
( ) |
|
Are the conclusions supported by the results? |
(x) |
( ) |
( ) |
( ) |
Comments and Suggestions for Authors
The paper aimed to determine the presence of TTXs in French Bivalve Mollusks and bacteria, and to understand its putative sources.
In general, the manuscript is well written and organized. The manuscript has a small number of figures, and I believe some of the figures supplied in the supplementary material (such as Figs. S1 or S3, for example), could be moved to the main text.
Again, we thank reviewer 2 for his appreciation of our work and the possibility to further improve the manuscript.
Following the recommendations of Reviewer 2, both Figure S1 and S3 were added into the main text and all the figures and the supplementary figure renumbered accordingly.
It requires additional clarifications of several aspects:
Table 1 – regarding the amount of toxin reported in bacterial strains, I question the presentation of results: were pellets weighed to allow the expression of results in µg/kg, as in bivalves? No details were given in section 5.1.2 as to how the number of bacterial cells in each culture was counted (or weighed).
Subsection 5.1.2 was dedicated to the collection of samples, not their processing (extraction and analysis). In fact, information pertaining to sample preparation was provided line 505, in the subsection 5.4.1 (“Extraction of TTXs for Shellfish and Bacteria”) in the section 5.4. (“Optimized Extraction and Analysis”): “For the extraction of TTXs, glass beads (250 mg, 0.15 to 0.25 mm, VWR) and 1% aqueous acetic acid (0.5 mL) were added to 200 mg of shellfish (oyster or clam) or 50 mg of bacterial cell pellet”. This allowed us to report LOD and LOQ as µg TTX-eq/kg of bacteria (as seen in Table 1). And if some of the 122 bacterial strains screened had been found positive, results would have been provided in this unit as well.
Line 221 – I suggest removing the comment ‘unfortunately’. ‘Luckily’, coastal waters are not full of toxin-producing microorganisms, and as result, we can get some food from coastal waters (such as bivalves)!
Sorry about the confusion, we removed the word “unfortunately” in the revised manuscript.
Section 4- I enjoyed the detailed discussion on the null results obtained. I wonder the ‘Arcachon’ mystery (of positive mouse bioassays), that lasted for a few years, was not related to TTXs after all? The authors could comment on this episode here, as, on some occasions, explanations for certain phenomena are not easy to get.
Thanks for appreciating our discussion.
We would not like to make any reference to the Arcachon-event in the manuscript as we feel it is not appropriate since we did not analyze any samples from that period or place in the present study (we presume you refer to the events referred to in this report: https://archimer.ifremer.fr/doc/00014/12568/).
We would like to remind the reviewer that the “Arcachon-mystery” toxin was detected using the organic fraction of partitioned extracts prepared for the “lipophilic mouse bioassay”, and positive results derived from the organic (= lipophilic) fractions of shellfish extracts. However, due to the water-soluble character of TTX, TTX does not move to the organic phase in the partitioning of the lipophilic mouse bioassay, at least up to a concentration of 200 µg/kg whole flesh equivalent (Vlamis, A., P. Katikou, I. Rodriguez, V. Rey, A. Alfonso, A. Papazachariou, T. Zacharaki, A.M. Botana, and L.M. Botana. 2015. First detection of tetrodotoxin in Greek shellfish by UPLC-MS/MS potentially linked to the presence of the dinoflagellate Prorocentrum minimum. Toxins. 7:1779-1807). It is therefore highly unlikely that the Arcachon incident would have been related to TTX (or STX).
Section 5.1.2 / Table 4 - explain the abbreviations used, please.
We presume you refer to Table S4 as there is no Table 4. The abbreviations used both in the subsection 5.1.2 and in Table S4 are the names of the laboratory holding bacterial culture collections. We are not sure this would be helpful to the readers as these are French and not English names. Still, the explanation of the laboratory abbreviations were provided in Table S4 (in both French and English).
Line 454 – the supplier for the Zobell medium was not detailed.
While the Marine Broth medium was purchased, the Zobell medium was prepared according to the reference provided ([67]: Zobell, C.E. Studies on marine bacteria. I. The cultural requirements of heterotrophic aerobes. J Mar Res 1941, 4, 42-75).
Section 5.3.2 – the writing is confusing. The SPE procedure was used, but also not used (filtration was used in alternative). All of these experimental steps were mixed in a single sentence. Please clarify.
In the Materials and Methods section, the method development and the final method used were separated in two distinct sections (section 5.3 “Method Optimization” and 5.4 “Optimized Extraction and Analysis”, respectively).
In subsection 5.3.2, both SPE and filtration were tested separately and finally, only filtrations were used for the purification of oyster and bacterial matrices, as mentioned line 508-509. We clarified this paragraph by splitting the sentence to read:
“We tested the recovery with two different SPE cartridges: ENVI-carb (Supelco, Sigma-Aldrich, Saint-Quentin-Fallavier, France) and GCB (Phenomenex, Le Pecq, France) following the protocol of Turner, et al. [21] (except that the sample loaded was 1 mL). In parallel, we also tested filtration (3 kDa or 0.2 µm, Nanosep or Nanosep MF, Pall, Saint-Germain-En-Laye, France) instead of SPE”.
Section 5.3.3. – contains the description of the LC columns used. This was repeated in section 5.4.2. Please merge sections.
Again, this subsection is in the section “Method optimization” where we compared three analytical columns. While the final conditions used were provided in section 5.4 “Optimized Extraction and Analysis”. Therefore, we do not want to merge the two paragraphs. As only the dimension and supplier of the column were repeated, we considered that no modification was required.
Submission Date
05 October 2021
Date of this review
13 Oct 2021 11:04:44